# Evolution and Efficiency Assessment of Pesticide and Fertiliser Inputs to Cultivated Land in China

**DOI:** 10.3390/ijerph18073771

**Published:** 2021-04-04

**Authors:** Xuesong Zhan, Chaofeng Shao, Rong He, Rongguang Shi

**Affiliations:** 1National & Local Joint Engineering Research Center on Biomass Resource Utilization, College of Environmental Science and Engineering, Nankai University, Tianjin 300350, China; 2120190632@mail.nankai.edu.cn; 2Sichuan Academy of Environmental Policy and Planning, Chengdu 610041, China; heenhr@163.com; 3Agro-Environmental Protection Institute, Ministry of Agriculture and Rural Afairs, Tianjin 300071, China; winsomesky@163.com

**Keywords:** pesticide and fertiliser reduction, agricultural production efficiency, Logarithmic Mean Divisia Index model, cluster analysis, sustainable agricultural development

## Abstract

Excessive use of pesticides and fertilisers has been a key issue limiting sustainable agricultural development. China is a typical pesticide- and chemical-fertiliser-dependent agricultural production area. We have matched the target indicators related to sustainable agricultural development (SDG1 and SDG2) and analysed the gap between China and four developed countries in terms of fertiliser and pesticide use intensity and efficiency from 2002 to 2016. We have used an improved Logarithmic Mean Divisia Index model and cluster analysis to identify the factors and effects driving increased pesticide and fertiliser inputs in China, and we discuss the exploratory effects of different provinces in reducing pesticide and fertiliser application and increasing efficiency. The findings reveal that (1) China is a typical pesticide- and fertiliser-dependent agricultural production area. The average combined fertiliser application efficiency in China from 2002 to 2016 was only 28% of that of the Netherlands, and the country’s average combined pesticide application efficiency was only 35% of that of the USA. (2) The most important of the three main drivers of the increase in pesticide and fertiliser inputs in China is the value added of the primary industry, contributing 56% for the period 2007–2016. (3) Further analysis at the provincial level according to four types—high-intensity high-yield type, high-intensity low-yield type, low-intensity high-yield type, and low-intensity low-yield type—clarified the provinces that should be focused on at the national level in terms of pesticide and fertiliser application reduction and efficiency increase in the future.

## 1. Introduction

Population growth and increasing socio-economic development are driving the increasing demand for food. The issue of food is related directly to the livelihood of farmers and the subsistence of the population. Of the 17 sustainable development goals in the United Nations 2030 Agenda for Sustainable Development, “No poverty” and “Zero hunger” have been set to address persistent and emerging challenges facing humanity and the planet. China achieved a great victory of poverty eradication in 2020, and the next key challenges to be addressed are consolidating the results of poverty eradication and coordinating the mechanism of poverty eradication and rural revitalisation. Construction of the rural ecological environment is vital for the sustainable development of the countryside.

Fertilisers promote crop growth by artificially supplementing nutrients, such as nitrogen, phosphorus, and potassium, essential for plant growth to achieve stable and increased agricultural production. Pesticides both prevent and kill crop pests and diseases to effectively reduce agricultural losses and ensure the yield and quality of agricultural products. Agricultural production has become increasingly dependent on chemical fertilisers and pesticides to ensure sustainable and stable crop yields, and the phenomenon of excessive application of chemical fertilisers and pesticides has become increasingly serious and harmful. Long-term excessive application of chemical fertilisers tends to cause soil consolidation and acidification, in addition to nutrient structure disorders, reducing the fertility of the soil and causing a decline in crop yield and quality [1]; the production of chemical fertilisers depends on mineral resources such as coal, petroleum, and apatite, resulting both in depletion of such resources and in environmental pollution; and excessive application of chemical fertilisers also causes serious agricultural surface pollution. As an example, the annual loss of nitrogen fertiliser applied to land in China is 124.8 kg/hm^2^, and the annual loss of phosphorus fertiliser is 38.8 kg/hm^2^. The excessive application of nitrogen and phosphorus nutrients causes serious eutrophication of water bodies and nitrate and nitrite pollution of groundwater through surface runoff and infiltration into rivers, lakes, and groundwater, and these can endanger human health through certain exposure pathways [2]. Pesticides are major “tri-causal” substances, and their carcinogenic and mutagenic latency can last up to decades [3]. After pesticide application in the field, only 10–30% of the applied amount can really be effective for crops, 20–30% will enter the atmosphere and water bodies through diffusion and runoff, and 50–60% will remain in the soil. Excessive application of pesticides represents a serious threat to farmers’ health, crop safety, and the ecological environment [4].

In 2015, as a response to the problem of pesticide and fertiliser pollution, the Ministry of Agriculture proposed a “zero-growth action on the use of pesticides and fertilisers by 2020”, requiring an increase in the rate of utilisation of pesticides and fertilisers, reducing the amount of pesticides and fertilisers applied. As of the end of 2020, after five years of implementation of this strategy, China has achieved its desired goal with respect to fertiliser and pesticide reduction and efficiency; fertiliser and pesticide use has been reduced significantly, the rate of utilisation of fertilisers and pesticides has been improved significantly, and high-quality development of the planting industry has been promoted. Although there have been remarkable achievements in China’s rural green development in recent years, the long-term extensive growth also makes China a typical global agricultural production area dependent on pesticides and fertilisers, and there is still a huge gap between China and developed countries in the control of pesticides and fertilisers.

This paper presents a comparative study on the intensity and efficiency of fertiliser and pesticide application in China, the United Kingdom (UK), France, the United States (USA), and the Netherlands based on data provided by the Food and Agriculture Organisation of the United Nations (FAO) on fertiliser application and grain production in each of these countries from 2002 to 2016. In addition, it explores the changing patterns of fertiliser and pesticide application intensity (PI) and efficiency in China. We analyse their effects on total fertiliser and pesticide application in China of three influencing factors: the value added of primary industry, the scale of rural operation, and the producer price of agricultural products. Furthermore, based on China’s rural statistics, we apply cluster analysis to pesticide and fertiliser application intensity (FI) and the value added of primary industry in 31 provinces in China (excluding Hong Kong, Macao, and Taiwan) and combine international experience to promote the high-quality development of agriculture in each part of China according to local conditions [5,6]. Through analysing the achievements made and problems faced by China with respect to agricultural development, this paper provides valuable experience for less-developed regions to solve food problems and agricultural pollution caused by the unreasonable use of pesticides and chemical fertilisers. The aim of the paper is to promote the realisation of the two sustainable development goals of “No poverty” and “Zero hunger” set by the United Nations 2030 Agenda for Sustainable Development.

## 2. Literature Review

In the context of the national advocacy of zero-growth and reduction in fertiliser and pesticide use, it has become vital to improve the efficiency of farmers’ use of fertilisers and pesticides. In studying spatial and temporal differences in pesticide and fertiliser application, Pingping Wang et al. [7] applied stochastic frontier analysis (SFA) to provincial panel data of 28 provinces (cities and districts) from 1991 to 2017 to measure the technical efficiency of fertiliser application in each province (city and district). Using provincial panel data from 1979 to 2015, Zhu Honghui et al. [8] measured and analysed the intensity and change trend of fertiliser application in China’s agriculture. Cai Rong et al. [9] used three SFA models to measure the agricultural fertiliser input efficiency in China from 1998 to 2017 on the basis of constructing a fertiliser consumption demand model, and they used panel data state transformation analysis to predict the dynamic trend of fertiliser input efficiency. Feng Tan et al. [10] applied stochastic frontier production function analysis to inter-provincial panel data in China from 2002 to 2012 to study the efficiency of pesticide application in each region of China and to carry out regional comparison and influence factor analysis. In recent years, increasing numbers of scholars have measured the efficiency of agricultural fertiliser inputs in China, using methods such as data envelopment analysis DEA [11,12] and SFA [13,14,15].

To analyse factors influencing pesticide application efficiency, Wang Yijie and Yang Zonghui et al. used comparative study analysis, the SDM model, the Tobit model, the C-D production function, and the grey correlation model to measure the influence of the value added of primary industry on utilisation of fertilisers and pesticides [16,17,18,19,20,21]. Shi Chang Liang and Tian Yun et al. used regression analysis, the Shapley value decomposition method, a rural household survey, and an Ordered Probit model to explore the influence of rural operation scale on fertiliser and pesticide use [22,23,24,25,26,27,28,29,30,31], and Wang Chunxiao and Ma Xiaoxing et al. studied the influence of agricultural price on pesticide and fertiliser use [32,33,34,35] (See Appendix A for details).

To summarise, scholars have conducted much research into pesticide and fertiliser application in rural China, with most of the study areas being provincial or prefecture-level cities; however, there has been relatively insufficient analysis of the temporal trends and influencing factors of fertiliser and pesticide inputs and outputs at the national level. To provide a basis for promoting high-quality agricultural development in various parts of China, this study utilises data from 2002 to 2016 to examine the changing trends of fertiliser and pesticide input intensity and efficiency in China, applies a comprehensive fertiliser and pesticide utilisation efficiency calculation model to comparison with some developed countries, and uses an improved Logarithmic Mean Divisia Index (LMDI) model and cluster analysis to identify the driving factors and driving effects of increased pesticide and fertiliser inputs in China.

## 3. Methods

### 3.1. Data Sources

Total pesticide use, pesticide use intensity, pesticide output rate (PO), and grain production per unit area in China and typical developed countries come from the FAO database; total fertiliser use in China and typical developed countries comes from the IFA database; and FI in typical countries is based on the fertiliser output rate (FO) from the World Bank statistics database and the FAO database. The data on the intensity of fertiliser use and the intensity of pesticide use in China’s provinces come from the National Bureau of Statistics; the overall producer price index for agricultural products is taken from the China Agricultural Product Price Survey Yearbook; and the value added of primary industry is taken from the Statistical Bulletin of the People’s Republic of China on National Economic and Social Development. The data on average sown area per household come primarily from the China Rural Statistical Yearbook and the China Population and Employment Statistical Yearbook. The domestic data in this paper are based on the provincial administrative regions of China. The agricultural land systems in Hong Kong, Macau, and Taiwan are not included because of certain differences from the mainland. The data cover the period of 1997–2016 for all 31 provinces (municipalities and autonomous regions) in China.

### 3.2. Research Methods

#### 3.2.1. Evolution of Pesticide and Fertiliser Application Efficiency in China and International Comparison

Despite remarkable achievements in China’s rural green development in recent years, the long-term crude growth has made China a typical agricultural production area dependent on pesticides and chemical fertilisers, and there is still a huge gap between China and developed counties in the control of pesticides and chemical fertilisers. This paper applies the calculation model of relative output rate and comprehensive efficiency of fertiliser proposed by Liu Qinpu et al. [36] to data provided by the Food and Agriculture Organisation of the United Nations on fertiliser application and grain production in each country from 2002 to 2016. We conducted a comparative study of the intensity and efficiency of fertiliser and pesticide application in China and the UK, France, the US, and the Netherlands to explore not only the changing patterns of fertiliser and land-use efficiency in China and major European and American countries but also the problems of fertiliser and pesticide application in China [37].

(1) Fertiliser and pesticide application intensity

*FI* refers to the amount of fertiliser applied per unit of crop sown area, and it is an important indicator in measuring the efficiency of fertiliser utilisation in a country. The calculation equation is
(1)$$FI=\frac{M}{A}$$
where *FI* denotes fertiliser application intensity, kg/hm^2^; *M* denotes total fertiliser application, kg; and *A* denotes crop sown area, hm^2^.

*PI* refers to the amount of pesticides applied per unit of crop sown area, and it is an important indicator in measuring the efficiency of pesticide use in a country. The calculation equation is
(2)$$PI=\frac{N}{A}$$
where *PI* indicates the intensity of pesticide application, kg/hm^2^; *N* indicates the total amount of pesticide application, kg; *A* indicates the crop sown area, hm^2^.

(2) Relative output rate of fertiliser and pesticide application

The *FO* refers to the yield of agricultural products per unit of fertiliser input and is computationally equal to the grain yield divided by the *FI*. As *FO* is a relative measure of intensity with a quantitative scale, it has no maximum upper limit, and it is inconvenient to make a typology of either high or low efficiency. To facilitate comparison of fertiliser application efficiency in different countries, with reference to the DEA method for measuring relative efficiency [21], the concept of Fertiliser Application Relative Output Rate (*FRO*), which is the ratio of a given *FO* to the maximum *FO*, has been proposed. The calculation equation is as follows:(3)$$FRO=\frac{FO}{FO_{MAX}}$$
(4)$$FO=\frac{Y}{FI}$$
where *FRO* denotes the relative output rate or net efficiency of fertiliser application; *FO* denotes the fertiliser application output rate, *FO_MAX_* denotes the maximum fertiliser application output rate, kg/kg; and *Y* denotes the crop yield, kg/hm^2^.

The *PO* refers to the yield of agricultural products corresponding to each unit of pesticide input, and the concept of relative output rate of pesticide application (*PRO*) is proposed in the same way, that is, the ratio of the output rate of a certain pesticide application to the maximum *PO*. The calculation equation is as follows:(5)$$\;PRO=\frac{PO}{PO_{MAX}}$$
(6)$$PO=\frac{Y}{PI}$$

(3) Yield scale factor

To reflect the crop yield scale of fertiliser and *PO*s, we propose the concept of yield scale coefficient (*YS*), which is the ratio of the yield of a certain unit of sown area of a crop to the yield of the highest unit of sown area, reflecting the scale effect of crop production. The larger the *YS*, the higher the crop yield indicated. The calculation equation is
(7)$$YS=\frac{Y}{Y_{MAX}}$$
where *YS* denotes the yield scale factor, and *Y_MAX_* denotes the maximum yield, kg/hm^2^.

(4) Efficiency of integrated application of fertilisers and pesticides

To reflect the scale effect of fertiliser application, the concept of Fertilisation Integrated Efficiency (*FIE*) is proposed as the open square of the product of the crop yield scale factor and the relative output rate of fertiliser application, calculated as
(8)$$FIE=\sqrt{YS\times FRO}$$
where *FIE* indicates the integrated relative output rate of fertiliser application, i.e., the integrated efficiency, and the larger the integrated efficiency, the better the combined effect of the *FRO* and crop yield.

#### 3.2.2. Factor Decomposition Analysis

This paper adopts the LMDI method [38] to factorise the total fertiliser and pesticide use in China; this method has been used widely because of its many excellent features. The LMDI method can not only eliminate the unexplained residual terms but also deal with the problem of zero values in the data, and it has desirable characteristics such as a simple calculation process and intuitive decomposition results. The LMDI method can make the model results more convincing. By compiling the existing studies in the literature (see Appendix A, we selected the value added of the primary industry, the scale of rural operations, and the total producer price index of agricultural products as the three decomposition factors of total fertiliser and pesticide use [39].

(i) Influence of primary industry value added on total fertiliser and pesticide use.

The application of pesticides and chemical fertilisers can boost crop yields and reduce farmers’ agricultural capital investment. In most provincial areas of China, there is a correlation between grain production and fertiliser application. Grain production is very highly dependent on fertiliser application, which is increasing year by year, and the phenomenon of random and additive fertiliser application is widespread in China [40]. The application of pesticides can easily and quickly reduce the damage to crops from pests and diseases.

(ii) Impact of the scale of rural operations on total fertiliser and pesticide use.

Research shows that the coexistence of different farming operation scales is a long-standing social phenomenon in China, and different scales of farming operation affect farming behaviour, such as production inputs, labour allocation, technology adoption, and planting motivation of farming households. The scale of rural operations plays a key role in the sustainable development of agriculture, and it has a profound impact on the economic and environmental performance of agricultural production [41]. In general, the larger the scale of the rural operation, the higher the level of machinery used. This facilitates the application of precision fertilisation techniques and management based on scientific knowledge, and the scale of the operation is more dependent on income from farmland and, thus, more sensitive to increases in fertiliser prices compared to the case with small farmers [42]. At the same time, the larger the scale of the operation, the more urgent the need for farmers to improve crop production through application of fertilisers and pesticides to provide nutrients for crops and kill pests, respectively, which may motivate farmers to increase their use. Therefore, there is a need to empirically test the impact of the scale of rural operations.

(iii) Influence of total agricultural producer price index on total fertiliser and pesticide use.

Agricultural product prices are an important component of the agricultural market. Price fluctuations of agricultural product affect not only the overall operation of the agricultural economy but also all aspects of agricultural production. Food output is directly related to farmers’ production investment. Farmers play a pivotal role in production investment. Price fluctuations of agricultural products as a signal of market changes influence farmers’ production investment decisions. According to the rational man assumption, farmers are profit maximisers in an ideal market model. When the prices of production factors remain constant, an increase in the price of agricultural products will cause an increase in expected returns, and farmers will tend to increase factor inputs to the point where marginal cost equals marginal return. Fertilisers and pesticides are important factors of production, and when the price of agricultural products falls, farmers may reduce fertiliser and pesticide inputs to obtain a relatively high net profit. The increase in agricultural product prices may also cause an increase in inputs of materials such as fertilisers and pesticides for farmers. In the current context of steadily increasing agricultural product prices, China’s agricultural policy orientation has shifted from increasing production to focusing on greenness and quality, and reduction in fertiliser use is an inevitable requirement for green, healthy, and sustainable agricultural development [43]. Consequently, there is a need to both clarify the relationship between the prices of agricultural products and fertiliser use and guide farmers to reduce their use of fertiliser on the basis of ensuring stable growth of farmers’ incomes.

We used the LMDI model to process the data and decompose the total fertiliser and pesticide use into three factors: the value added of primary industry, the scale of rural operation, and the total producer price index of agricultural products. The LMDI calculates the specific influence and contribution of each factor to either the increase or the decrease in use of fertilisers and pesticides, and the results can then be analysed to identify the main factors influencing the total fertiliser and pesticide use. The additive equation of LMDI is expressed as follows:ΔF = F_t_ − F_0_ = ΔF_GDP-pi_ + ΔF_TPPPI-ap_ + ΔF_FLCI_(9)
where F_t_ and F_0_ denote the total fertiliser–pesticide use in period t and the base period, respectively; ΔF_GDP-pi_ represents the change in total fertiliser and pesticide use due to value added in the primary sector; ΔF_TPPPI-ap_ represents the change in total fertiliser and pesticide use due to agricultural producer prices; and ΔF*FLCI* represents the change in total fertiliser and pesticide use due to the scale of rural operations. Table 1 presents the equations for each decomposition parameter. 

#### 3.2.3. Cluster Analysis

The results of the factor decomposition analysis reveal that the value added of the primary industry has the most significant impact on fertiliser and pesticide use. To understand the characteristics of the relationship between the intensity of fertiliser and pesticide application and crop yield in each province, this paper presents the intensity of fertiliser and pesticide use and the value added of primary industry in each province in recent years. First, the 31 provinces have been classified into the following four categories: high-intensity, high-yield (HS–HY); high-intensity, low-yield (HS–LY); low-intensity, high-yield (LS–HY); and low-intensity, low-yield (LS–LY), using the intensity of pesticide use as the intensity indicator and the value added of primary industry as the yield indicator.

Then, the intensity of fertiliser and pesticide application in each province was equated to a horizontal coordinate, the value added of primary industry in each province was equated to a vertical coordinate, and the results were arranged in descending order. The distance between each point and the top of the histogram shows the combination of intensity–yield characteristics. Lastly, we used the four provinces closest to the fixed points of the histogram as the initial clustering centres and the Euclidean distance for clustering analysis. The different provinces have been classified within the four different intensity–yield categories [44].

## 4. Results

### 4.1. Evolution of Pesticide and Fertiliser Application Efficiency in China and International Comparison

#### 4.1.1. Evolution of Pesticide and Chemical FI and International Comparison

Figure 1 shows that the intensity of fertiliser use in China rose steadily until 2015, when the Ministry of Agriculture launched a zero-growth action to increase the efficiency of and promote the reduction in fertiliser and pesticide use. This was the first time that the intensity of fertiliser use in China decreased. However, fertiliser use in China is still far higher than that in the US, France, the Netherlands, the UK, and other countries. Intensity of fertiliser application in China in 2015 was 506.1 kg/hm^2^, which was three times that of France, two times that of the Netherlands, two times that of the UK, and 3.7 times that of the US, respectively, and far greater than the global average fertiliser use of 138.9 kg/hm^2^.

Figure 2 shows that pesticide use has been increasing rapidly in China, and this growth has been accompanied by an increase in intensity of pesticide use (pesticide use per unit of crop sown area), which declined for the first time only in 2014. In 2015, the Ministry of Agriculture launched a zero-growth action against use of chemical fertilisers and pesticides, which reduced chemical fertiliser and pesticide use and increased efficiency. Over the past 20 years, the intensity of pesticide use in China has grown by 42.4%. Over the same period, the intensity of pesticide use in the US, the UK, France, and other countries has been declining, and it remains at a low level. Before 2001, the intensity of pesticide use in the Netherlands was higher than that in China; however, the intensity of pesticide use in the Netherlands has been decreasing since the implementation of the new policy on pesticide in 1990, and the intensity of pesticide use in the Netherlands was already 4.45 kg/hm^2^ lower than that in China by 2016. To reduce the impact of pesticide use on the ecological environment, France implemented a pesticide reduction plan in 2008, with the goal of reducing pesticide use by 50% over 10 years. The main types of pesticides used in the UK are herbicides; before 2005, the annual use of herbicides in the UK was approximately 20–25 thousand tonnes, whereas, after 2005, the use of herbicides decreased significantly due to the use of alternative herbicides with higher activity at lower dosage. FAO data show that the intensity of pesticide use in China in 2016 reached 13.06 kg/hm^2^, ranking 15th in the world and higher than in developed countries such as the US, the UK, France, and The Netherlands.

#### 4.1.2. Evolution of Pesticide and Fertiliser Application Efficiency in China and International Comparison

Based on Equations (2) and (3), we have calculated the relative output rates of fertiliser application, i.e., net efficiency, for China and four other countries from 2002 to 2016. China’s total fertiliser use and intensity of use were both at a high international level, but the relative FO was low, even far below those of the developed countries and the world average.

The relative output rate of fertiliser application only reflects the net rate of fertiliser utilisation of a country. However, agricultural production that aspires to a high yield per unit area requires a high rate of fertiliser utilisation, i.e., high efficiency and high output, which better reflects the meaning and role of fertiliser inputs. Therefore, considering the combined effects of relative FO and grain yield, we used Equations (4) and (5) to calculate the combined efficiency of fertiliser application in each country for each year (Figure 3). The larger the combined efficiency, the more efficient the fertiliser application. From 2002 to 2016, apart from 2011, the combined efficiency of fertiliser application from the largest to the smallest was ranked as follows: the Netherlands, the US, France, the UK, and China, the average values of which were 0.95, 0.66, 0.58, 0.50, and 0.27, respectively (Table 2). It can be said that the overall efficiency of fertiliser application better reflects the actual situation of fertiliser utilisation in each country.

From 1997 to 2016, the PO in China fell from 525.3 kg/kg to 461.7 kg/kg, a reduction of 63.6 kg/kg, whereas the POs of developed countries such as France, the UK, the US, and the Netherlands increased by 309.2 kg/kg, 1002.3 kg/kg, and 907.3 kg/kg, respectively. Overall, the POs of China and the Netherlands were significantly lower than those of other countries, mainly due to the higher intensity of pesticide use in both countries (Figure 4). The larger the combined efficiency, the more efficient the pesticide application. The combined efficiency of pesticide application from the largest to the smallest was ranked as follows: France, the Netherlands, the US, China and the UK, the average values of which were 0.69, 0.68, 0.48, 0.28, and 0.28, respectively (Table 3).

### 4.2. Analysis of the Driving Effects of Pesticide and Fertiliser Inputs in China

We used the LMDI model to process the data and decompose the total fertiliser and pesticide use into three factors: the value added of the primary industry, the scale of rural operations, and the total producer price index of agricultural products. In addition, we calculated the specific impact and rate of contribution each factor to either the increase or the decrease in fertiliser and pesticide use (Table 4), and then we analysed the results to identify the main factors affecting the total fertiliser and pesticide use.

We divided the 1997–2016 time series into two phases: 1997–2006 and 2007–2016. We used the equations to calculate the contribution value and contribution rate (in which the contribution rate is equal to the ratio of the absolute value of the contribution of the effect to the sum of the absolute value of the contribution of the effect) of the following three factors: primary industry value added, rural operation scale, and the total agricultural producer price index for the total fertiliser and pesticide use.

As the factors are in a state of fluctuation, the contribution of each effect to the total pesticide use is expressed more visually through horizontal bar charts, and the contribution of each factor is shown in Figure 5 and Figure 6.

The histogram of the contribution of the factors plotted reveals that the contribution of the value added of the primary industry to the total fertiliser and pesticide use from 1997 to 2006 was more than 100%. There was a significant downward trend in the degree of scale of arable land during this period, and the total agricultural producer price index first declined and then showed a significant upward trend after 2002. There was a significant upward trend in both the degree of scale of arable land and the total agricultural producer price index from 2007 to 2016. The degree of influence of two factors—the degree of scale of arable land and the total price index of agricultural products—on the total use of fertilisers and pesticides increased significantly from 2007 to 2016. The influence rate of the added value of primary industry on the total fertiliser and pesticide use exceeded 50%. As a developing country, China possesses huge potential for economic development; between 1997 and 2016, the growth rate of the value added of primary industry in China reached 75.99%. The contribution of the value added of the primary industry to the total fertiliser and pesticide use is anticipated to increase further. Though scale of operation of arable land facilitates the management of fertiliser and pesticide use, the current rural land transfer is still fragmented, the land transfer rate is still low, and the scale development trend has not yet been formed. Consequently, the impact of the scale level on the intensity of fertiliser and pesticide use is limited.

### 4.3. Cluster Analysis Based on the Intensity of Pesticide and Fertiliser Use and the Value Added of Primary Industry in 2016 by Provinces in China

The LMDI model analysis shows that the value added of primary industry in China had the greatest impact on the total use of fertilisers and pesticides. There was an increase in the degree of impact of the scale of rural operations and producer prices of agricultural products on the total amount of use. The above three influencing factors play a crucial role in promoting both reduction and efficiency of chemical fertiliser, pesticide reduction, and pest control in China, thereby determining whether China can achieve transformational development and high-quality development and follow a modern agricultural development path with high output efficiency, product safety, resource conservation, and environmental friendliness. However, China’s regional development is uneven, with large disparities between provinces and regions with respect to natural geographic conditions, and the performance characteristics and influence effects of the three influencing factors of value added of primary industries, rural business scale, and producer prices of agricultural products differ. Therefore, based on China’s rural statistics, we used the cluster analysis method to analyse the intensity of pesticide and fertiliser use and the value added of the primary industry in 31 Chinese provinces (apart from Hong Kong, Macao, and Taiwan), and we combined international experience to promote the high-quality development of agriculture in various parts of China according to local conditions.

#### 4.3.1. Cluster Analysis Based on the Intensity of Pesticide Use and Value Added of Primary Industry in 2016 by Provinces in China

Using the intensity of pesticide use as an indicator of intensity and the value added of primary industry as an indicator of production, we classified the 31 provinces into the four categories of HI–HY, HI–LY, LI–HY, and LI–LY. The horizontal coordinate is the intensity of pesticide use in each province, and the vertical coordinate is the value added of primary industry in each province, with the coordinates arranged in descending order to form the histogram in Figure 7. The distance between each point and the vertex of the histogram shows the combination of “intensity–yield” characteristics.

We used the provinces of Henan, Guangdong, Ningxia, and Hainan, those with the shortest distance from the histogram’s vertices, as the initial clustering centres and then standardised the data and clustered them by Euclidean distance (Table 5).

#### 4.3.2. Cluster Analysis Based on the Intensity of Fertiliser Use and the Value Added by Primary Industry in Each Province of China in 2016

Using the fertiliser use intensity as an indicator of intensity and the value added of primary industry as an indicator of production, we classified the 31 provinces into the four categories of HI–HY, HI–LY, LI–HY, and LI–LY. The horizontal coordinate is the intensity of fertiliser use in each province, and the vertical coordinate is the value added of primary industry in each province, with the coordinates arranged in descending order to form the histogram in Figure 8. The distance between each point and the top of the histogram shows the combination of “intensity–yield” characteristics.

We used the provinces of Shandong, Beijing, Sichuan, and Qinghai, those with the shortest distance from the histogram vertices, as the initial clustering centres, and then we standardised the data and clustered them by Euclidean distance (Table 6).

#### 4.3.3. Overall Evaluation of the Intensity of Fertiliser and Pesticide Use and Cluster Analysis of the Value Added of the Primary Sector

The results of the cluster analysis of FI and the value added of primary industry against the results of the cluster analysis of pesticide application intensity and value added of primary industry in 2016 revealed that 16 of the 31 provinces had the same type of “intensity–yield” of fertilisers and pesticides. Table 7 shows the cluster analysis results for 16 provinces.

## 5. Discussion

In this paper, we focus on two core tasks. The first task is a comparative study of the trends in fertiliser and pesticide use intensity and integrated application efficiency in China and four developed countries (UK, USA, France, and The Netherlands); the second task involves using an improved LMDI model and cluster analysis to identify the driving factors and driving effects of increased pesticide and chemical fertiliser inputs in China and to discuss the exploratory effects of different provinces in reducing the use of chemical fertilisers and pesticides and in increasing efficiency based on the results of the impact factor analysis.

### 5.1. Evolution of Pesticide and Fertiliser Application Efficiency in China and International Comparison

China’s consumption of pesticides and fertilisers ranks highest in the world, and the country shows an overall trend of rapid growth until 2012, after which the rate of growth tends to level off. Compared with other countries, China’s total use and intensity of use of pesticides and fertilisers are both high, and its comprehensive efficiency of utilisation of pesticides and fertilisers is not high and is far below the world average. Overall, the gap between grain crop yields per unit area of grain in China and those in typical developed countries is narrowing gradually; however, there is significant variation in the quality of major grain varieties, and greater efforts are needed to close the gap.

Application efficiency is measured mainly by the relative output ratio of grain crop yield to application of fertilisers and pesticides. Cereal yields in the five countries studied exhibit a fluctuating upward trend, and the gap between China’s grain crop yields and those of typical developed countries per unit area of grain is narrowing gradually. However, in recent years, China’s grain production efforts and scientific research have focused mainly on increasing grain quantity, neglecting improvement in grain quality, which, in turn, has resulted in significant variation in the quality of major grain varieties. The world’s major developed countries and agricultural countries attach considerable importance to cultivating and promoting good crop varieties [45], and the US, France, and other countries have listed variety selection and promotion as a key support project of their national agricultural development plans. China should change its traditional “quantity, light quality” concept of food production and scientific and technological innovation and establish a new paradigm of food development with equal emphasis on quantity and quality. China should both increase significantly its yield of grain crops and improve the crop quality.

China’s total fertiliser and pesticide use and intensity of use are at a high international level; however, the country’s relative output rate is low, even far below those of the developed countries and the world average. In recent years, due to the implementation of measures to reduce and increase the efficiency of chemical fertilisers and pesticides in China, the country’s grain production has continued to increase, and there has been slight improvement in the relative output rate of application. Since 2015, the Ministry of Agriculture and Rural Affairs has deployed an in-depth zero-growth action of chemical fertilisers and pesticides to reduce their use and increase their efficiency to achieve significant results. Grain crops make up the largest proportion of the total sown area of crops. Between 2015 and 2019, there has been a fluctuating upward trend of China’s total sown area of crops. In 2019, the total sown area of crops China was 165,931 thousand hectares, an increase of 29 thousand hectares from the previous year, at a rate of increase of 0.02% year-on-year. The use of agricultural fertilisers is continuing to decline. In 2018, 565.342 million tonnes of agricultural fertilisers were applied in China, which was down by 2.0599 million tonnes from the previous year, at a rate of decrease of 3.52% year-on-year. In 2019, 54.0359 million tonnes of agricultural fertiliser were applied in China, which was down by 2.4983 million tonnes from the previous year, with a rate of decrease of 4.42% year-on-year. The use of pesticides has also shown a continuous downward trend, with 1.5 million tonnes of pesticides applied in 2018, down 9.64% year-on-year, and 1.39 million tonnes applied in 2019, down 7.33% year-on-year. Scientific calculations show that, in 2019, the fertiliser utilisation rate for the three major food crops of rice, corn, and wheat in China was 39.2%, 1.4 percentage points higher than in 2017, and 4.0 percentage points higher than in 2015. Meanwhile, the pesticide utilisation rate was 39.8%, 1.0 percentage points higher than in 2017, and 3.2 percentage points higher than in 2015 [46].

### 5.2. Analysis of the Driving Effects of Pesticide and Fertiliser Inputs in China

Analysis of the contributions of the three factors of the value added of the primary industry, the scale of rural operations, and the total producer price index of agricultural products has shown that the value added of the primary industry has a more significant impact on the use of fertilizers and pesticides [21]. The value added of the primary industry is anticipated to further increase the value of its contribution to the total fertiliser and pesticide use in the future. However, this effect shows a tendency to weaken [19]. Factors such as fertilisers and pesticides have the problem of real losses in terms of inputs; this is not conducive to an increase in food production value [20]. The scale of arable land facilitates the management of fertiliser and pesticide application [23], and the fine fragmentation of arable land has a significant, negative impact on the efficiency of farmers’ fertiliser use. This is an important reason for the low efficiency of farmers’ fertiliser use [13]. However, rural land transfer is still fragmented, and the rate of land transfer is still low and has not yet assumed a trend of large-scale development; consequently, the level of scale has limited impact on the intensity of use of fertilisers and pesticides [47]. Meanwhile, farmers with different scales of operation choose different fertilisers and pesticides from each other [24]. With the implementation of the national rural revitalisation strategy, the revised Rural Land Contract Law in 2018 and the Civil Code adopted in 2020 have moderately opened up rural land to shareholdings and mortgages, increased the provisions of land management rights, and confirmed the system of separation of the three rights of rural land, thereby providing effective institutional guarantees to promote large-scale agricultural operations in addition to rural land transfer in China [48]. The fluctuation of agricultural prices affects not only the overall operation of China’s agricultural economy but also all aspects of agricultural production [49]. The degree of influence of the two factors of the total producer price index of agricultural products on the total use of fertilisers and pesticides has been found to also show a significant upward trend, and positive correlation between the price of agricultural products and the application of fertilisers has been established [29].

### 5.3. Cluster Analysis Based on the Intensity of Pesticide and Fertiliser Use and the Value Added of Primary Industry in 2016 by Provinces in China

#### 5.3.1. Cluster Analysis Based on the Intensity of Pesticide Use and Value Added of Primary Industry in 2016 by Provinces in China

The provinces of Hainan and Shanghai are in the HI–LY category. Hainan is located in the agricultural and forestry area of Qionglei and the islands of the South China Sea. According to the 2019 National Arable Land Quality Grade Status Bulletin, Hainan is dominated by purple, limestone (rock), and sandy soils of relatively low basic strength, and it lacks farmland infrastructure [50]. Moreover, some arable land shows either acidification or salinity or is affected by other factors, resulting in Hainan’s primary industry value added being at a low level [51]. Meanwhile, as Hainan’s natural climatic conditions of high temperature and high humidity result in more severe occurrences of pests and diseases often than are seen inland, and farmers frequently use more pesticides to ensure the production of agricultural products.

Shanghai is located on the plains of the lower reaches of the Yangtze River, and it is a hilly agricultural and fishery area, with mainly yellowish-brown and brown soil, poor standing conditions, large topographic fluctuations, poor levels of soil nutrients, low basic strength, backward water conservancy facilities, inadequate irrigation conditions, and obstacles in some of the arable land [52]. Farmers in Shanghai apply a lot of pesticides to increase production, resulting in severe pesticide pollution.

#### 5.3.2. Cluster Analysis Based on the Intensity of Fertiliser Use and the Value Added by Primary Industry in Each Province of China in 2016

The results of the cluster analysis show that the intensity of fertiliser application is high in Beijing, Tianjin, Jilin, Shanghai, Fujian, Hainan, Shaanxi, and Xinjiang, though the added value of primary industry is relatively low. Beijing and Tianjin both face the problem of unbalanced fertiliser application structure, and over-fertilisation by farmers is the main cause of surface pollution by fertilisers.

A comparison of the total area planted with crops and the intensity of fertiliser application in Beijing and Tianjin over the past 20 years reveals a downward trend in the two regions; however, there has been an increase in the area planted with cash crops and vegetables. The intensity of fertiliser application is exhibiting a trend of continuous growth [53]. Additionally, urbanisation and industrialisation have caused a decline in the sown area of crops in the region, resulting in a continued, rapid increase in the intensity of chemical fertiliser application, thus making this region a key area for the prevention and control of surface pollution [54].

Although Jilin province is an important grain production base in China, the analysis reveals it to be in a low-yield region. The factors affecting grain production in Jilin include the following: first, Jilin’s farmland infrastructure is backward, with a large area of low- and medium-yielding fields and declining soil fertility; second, drought has become an important factor restricting agricultural development in this province; and third, with the shift of the agricultural labour force, there has been a decline in the quality of the labour force engaged in agricultural production. These problems have had a severe impact on food production in Jilin [55]. In 2007, Jilin province launched a project to increase its food production capacity by an additional 10 billion kilograms per year; this entailed using more chemical fertilisers to increase production, thus causing an increase in chemical fertiliser pollution [56].

Fujian province is located in the middle and lower reaches of the Yangtze River, in hilly and mountainous areas of low-quality arable land with poor standing conditions, large topographic fluctuations, poor soil nutrients, low basic strength, and poor field infrastructure. These limitations of the land affect the province’s food production. Nevertheless, Fujian is a large agricultural province. Since the start of the new millennium, agriculture in Fujian province has continued to develop rapidly; however, there has been no significant change in the extensive agricultural development pattern of “high input, high output, high emissions”, and the change in planting structure has exacerbated agricultural surface pollution. Although grain production in Fujian province has declined, there has been a rapid increase in the production of cash crops. Cash crops have a higher demand for chemicals such as fertilisers than do food crops [57]. Moreover, the increased area planted with cash crops has resulted in an increase in the number of fertiliser inputs required.

Shaanxi province is located in China’s inland hinterland. Though there are many people there, the little available arable land is of poor quality, reserve resources are inadequate, and the situation with regard to the protection of arable land is very serious. In recent years, conditions of Shaanxi agricultural production have improved significantly; however, neither the scale nor the intensification are high, the gap in food production has widened, the level of livestock production has lingered, the process of development of modern agriculture is slow, agricultural production generally has low yields, and labour efficiency is not high, all of which negatively affect agricultural production in Shaanxi province. As a traditional agricultural province in the west of China, the production and operational behaviour and the organisational system of farm households in Shaanxi province are typical of traditional farm households in China, combining traditional agriculture with a high degree of fertilisation, leading to excessive use of chemical fertilisers [58].

As a large agricultural province (region), Xinjiang’s food production occupies an important place in its economic structure. However, Xinjiang’s location in the southwestern part of China, deep inland, where it is hot all year round and there is little rainfall, has led to a shortage of surface water and poor agricultural irrigation conditions, which poses enormous challenges for food production in the province.

Agricultural development in Xinjiang is currently facing the problem of agricultural surface pollution, which restricts the sustainable development of agriculture. Fertiliser input source pollution is one of the main types of agricultural source pollution in Xinjiang, where the intensity of fertiliser application is far above safe standards for fertiliser use. The main focus is on application of nitrogen and phosphorus fertilisers [59]. Meanwhile, to increase production, farmers tend to over-apply fertilisers, resulting in high-intensity use of chemical fertilisers in agricultural production.

#### 5.3.3. Overall Evaluation of the Intensity of Fertiliser and Pesticide Use, and Cluster Analysis of the Value Added of the Primary Sector

China is a vast country, with huge differences between provinces in terms of natural geographic conditions and agricultural development, and the amount of chemical fertilisers and pesticides applied also differs between provinces. The results of the cluster analysis of the intensity of pesticide and fertiliser use and the value added of primary industries in each province of China in 2016 show that the provinces of Guangdong and Hubei are in the HI–HY category. As part of China’s main food supply area, they need to maintain production of agricultural products in accordance with an optimal pesticide and fertiliser reduction and efficiency technology model. Both Hainan and Shanghai are in the HI–LY group. Although the region has high intensity of application of chemical fertilisers and pesticides, the crop yield is low, and there is need to reduce the pressure on agricultural production, through the use of soil testing and fertilisation, planting structure adjustment, and other measures to reduce agricultural input. The five provinces of Hebei, Anhui, Hunan, Guangxi, and Sichuan are in the LI–HY category; they have low intensity of fertiliser and pesticide, coupled with high yield. The level of agricultural development is relatively high, the future offers good potential for agricultural development, and there is a need to improve the quality and efficiency of agriculture without increasing the inputs of pesticides and fertilisers. The four provinces of Yunnan, Tibet, Gansu, and Qinghai are in the LI–LY category and are key to improving the sustainable development of China’s agriculture. Aside from the above provinces, Fujian Province has a relatively high level of intensity of fertiliser and pesticide application, which should be adapted to the local conditions, and a pesticide and fertiliser reduction and efficiency improvement technology model suitable for the local geographic and cultural conditions should be sought.

In recent successive years, chemical fertilisers and pesticides have played a key role in China’s grain harvests; however, they have had certain impacts on the environment [60]. Before 2015, there was continuous growth in the total amount of chemical fertiliser and pesticide application in China, causing great challenges and pressures for the development of ecological green agriculture and for efforts for reduction and efficiency. Use of chemical fertilisers and pesticides in central, western, and north-eastern regions increased significantly, whereas the application of chemical fertilisers and pesticides declined somewhat in the eastern region, though the scale of application was still high. Of the 31 provinces studied, Shandong, Jiangsu, Henan, Hubei, and Guangdong belong to high fertiliser application, high-growth areas; Qinghai, Shanxi, Inner Mongolia, Jiangxi, Chongqing, Guizhou, Yunnan, Tibet, Gansu, and Ningxia belong to negative growth, low-fertiliser application areas; Guangdong, Liaoning, Zhejiang, Fujian, Jiangxi, and Hubei belong to high pesticide application and high-growth areas; and Ningxia, Beijing, Tianjin, Shanxi, Inner Mongolia, Jilin, Heilongjiang, Chongqing, Guizhou, Yunnan, Tibet, Shaanxi, Gansu, Qinghai, and Xinjiang belong to negative growth low-pesticide application areas.

Since the promotion of the “double reduction” programme for application of fertilisers and pesticides, both the total amount and the intensity of the fertilisers and pesticides used in agricultural have declined greatly throughout China; however, as total amount of agricultural fertilisers and pesticides used in China is so high, the intensity of use is still too high, and there is still enormous room for improvement in reducing fertiliser and pesticide use, particularly in areas with high pesticide and fertiliser input intensity and low value added in primary industries, strengthening the policy of double reduction of chemical fertilizers and pesticides is particularly critical [61].

## 6. Conclusions

The intensity of pesticide and fertiliser application in China is high and has been increasing steadily. However, the comprehensive efficiency of utilisation of pesticides and fertilisers in China is not high and is far below the world average. The average comprehensive efficiency of application of fertilisers in China from 2002 to 2016 was 0.27, which was only 28% of the comprehensive efficiency of application in the Netherlands and 41% of that in the US. The average comprehensive efficiency of application pesticides in China was 0.28, which is only 35% of that of the USA and 40% of that of the UK. Of the three factors of value added of primary industry, scale of rural operation, and producer price of agricultural products, value added of primary industry has the greatest influence on total fertiliser and pesticide application. The degree of influence of the scale of rural operations and producer prices of agricultural products on the total use has increased. The value added of the primary industry contributed 56% from 2007 to 2016, and the total agricultural producer price index and the scale of rural operations contributed 35.51% and 8.49%, respectively. The results of cluster analysis of the intensity of pesticide and fertiliser use and the value added of the primary industry in 2016 in 31 provinces showed that Hainan and Shanghai had high intensity of pesticide application but low value added of the primary industry, and Beijing, Tianjin, Jilin, Shanghai, Fujian, Hainan, Shaanxi, and Xinjiang had high intensity of fertiliser application but relatively low value added of the primary industry. There is huge and imperative pressure to reduce the application of fertilisers and pesticides.

As of the end of 2020, China has successfully achieved the expected goals in fertiliser and pesticide reduction and efficiency, with a significant reduction in the application of fertilisers and pesticides and a significant increase in the efficiency of fertilisers and pesticides, thereby promoting the high-quality development of the plantation industry, with obvious results. However, the agricultural surface pollution resulting from the excessive use of pesticides and chemical fertilisers has become a bottleneck for the sustainable development of China’s agriculture. In the future, we will conduct further research work through questionnaires and field interviews to guide, in a more scientific and effective manner, the actions toward pesticide and chemical fertiliser reduction and increased efficiency for different crops, in addition to farming methods and planting structures in various regions, promote agricultural transformation and upgrading, follow the route of socialist rural revitalisation with Chinese characteristics, and accelerate the modernisation of both agriculture and rural areas.

## Figures and Tables

**Figure 1 ijerph-18-03771-f001:**
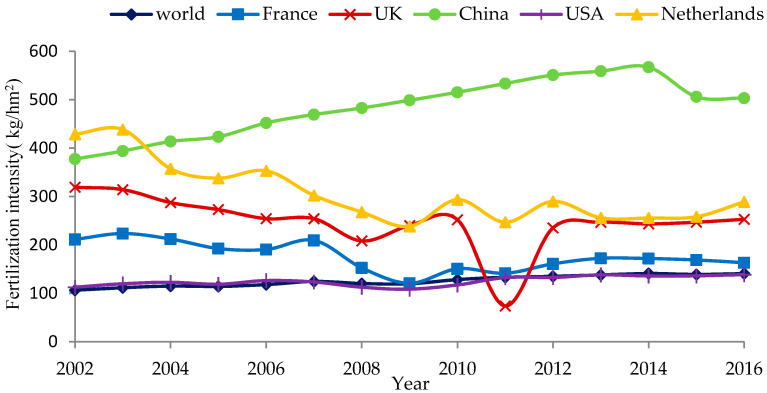
Comparison of intensity of fertiliser application in typical developed countries.

**Figure 2 ijerph-18-03771-f002:**
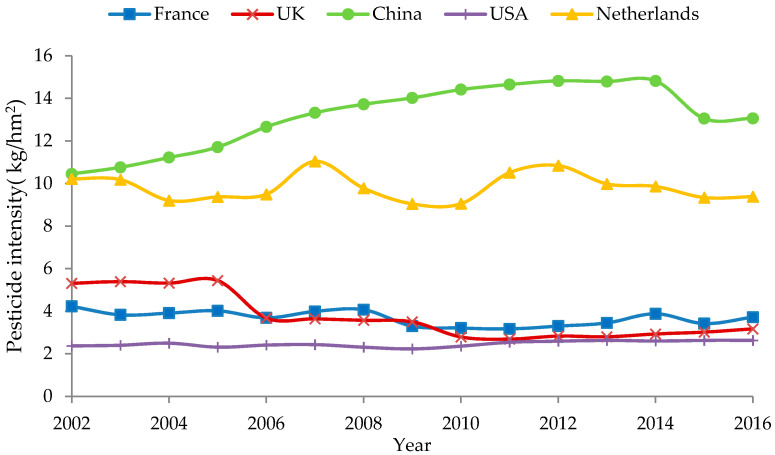
Comparison of pesticide use intensity in typical developed countries.

**Figure 3 ijerph-18-03771-f003:**
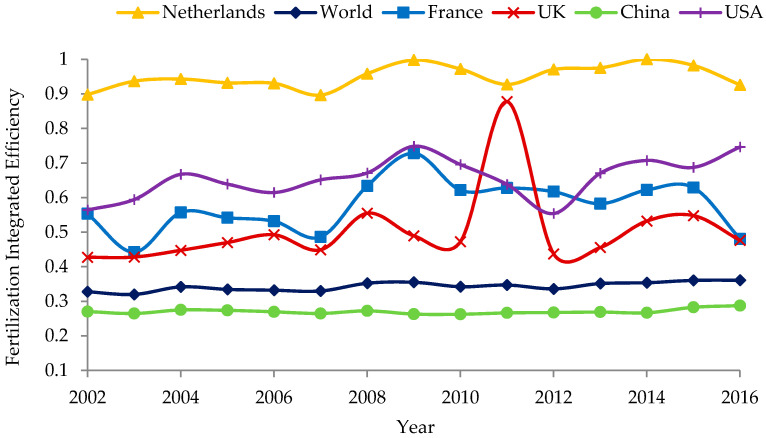
Comparison of the Fertilisation Integrated Efficiency (FIE) in typical countries.

**Figure 4 ijerph-18-03771-f004:**
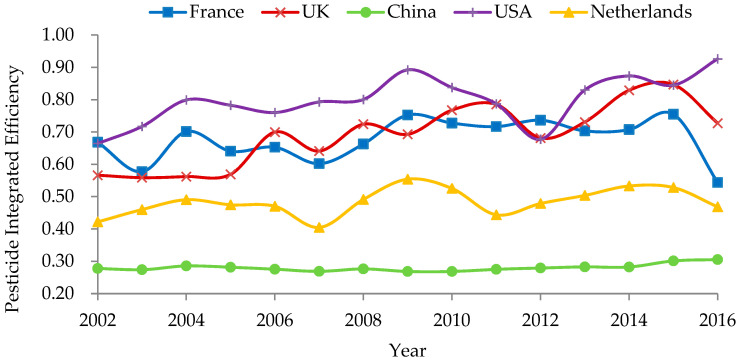
Comparison of pesticide integrated efficiency output rate in typical countries.

**Figure 5 ijerph-18-03771-f005:**
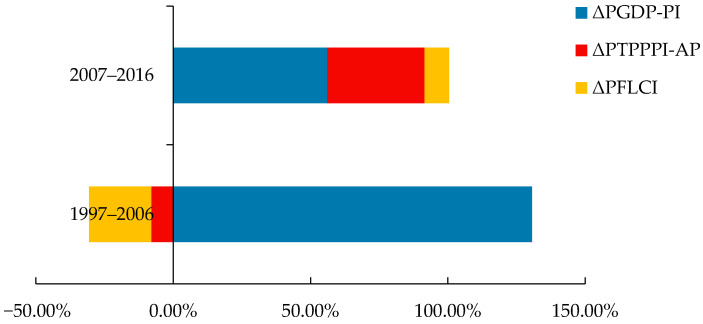
Contribution rate of influencing factors of total pesticide use.

**Figure 6 ijerph-18-03771-f006:**
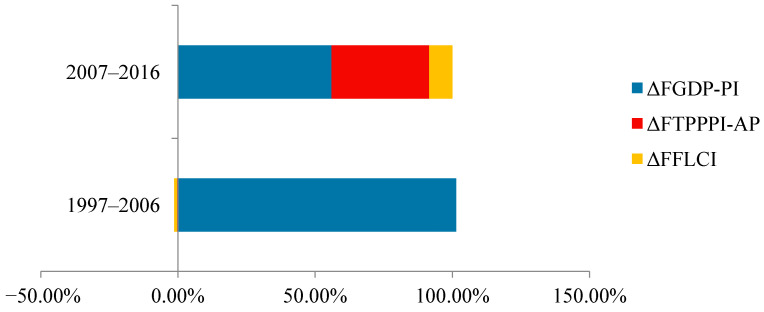
Contribution rate of influencing factors of total fertiliser use.

**Figure 7 ijerph-18-03771-f007:**
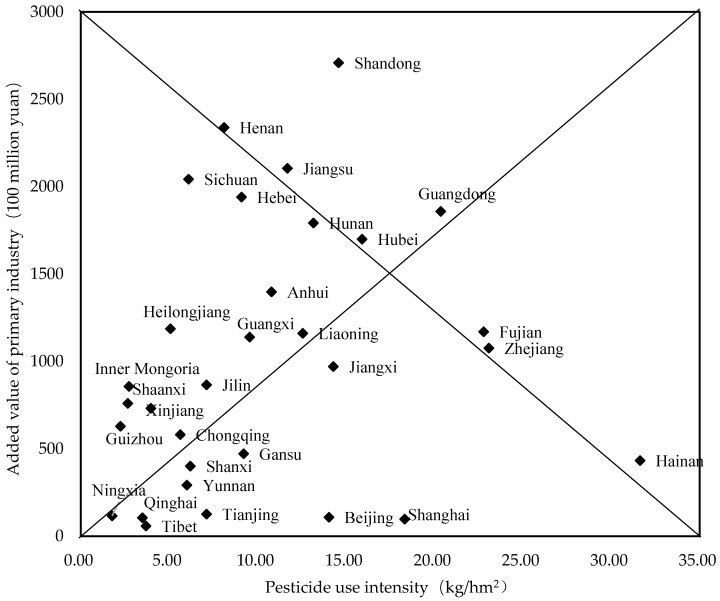
Annual average pesticide use intensity and primary industry added value in China’s provinces in 2016.

**Figure 8 ijerph-18-03771-f008:**
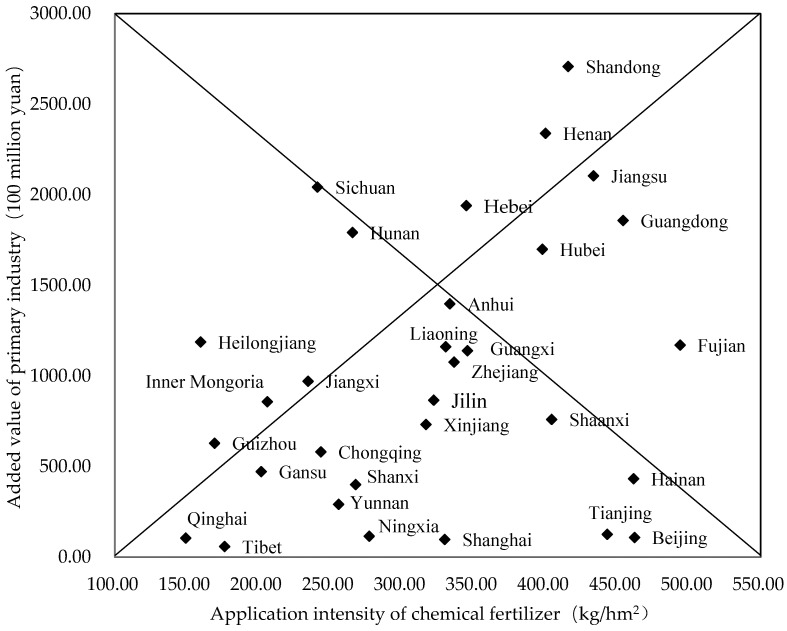
Annual average fertiliser use intensity and added value of primary industry in Chinese provinces in 2016.

**Table 1 ijerph-18-03771-t001:** Decomposition parameters of LMDI.

Factor	Variable	LMDI Equation
Value added of primary industry	GDP-pi	$\Delta F_{GDP-pi}=\frac{FT-F0}{lnFT-lnF0}ln\left(\frac{GT}{G0}\right)$
Agricultural producer price index	TPPPI-ap	$\Delta F_{TPPPI-ap}=\frac{FT-F0}{lnFT-lnF0}ln\left(\frac{TT}{T0}\right)$
Scale of rural operations	FLCI	$\Delta F_{FLCI}=\frac{FT-F0}{lnFT-lnF0}ln\left(\frac{LT}{L0}\right)$

Note: G = GDP-pi, T = TPPPI-ap, L = FLCI.

**Table 2 ijerph-18-03771-t002:** Fertilisation intensity, grain yield, relative output rate, and integrated efficiency of each country based on multi-year average in 2002–2016.

Country Name	Fertilisation Intensity(kg/hm^2^)	Cereal Yield(kg/hm^2^)	Fertilisation Relative Output	Fertilisation Integrated Efficiency
World	125.53	3562.97	0.39	0.34
France	176.03	7012.57	0.77	0.58
UK	246.67	7078.87	0.78	0.5
China	483.15	5500.07	0.61	0.27
USA	124.94	6800.61	0.75	0.66
Netherlands	307.45	8191.63	0.9	0.95

**Table 3 ijerph-18-03771-t003:** Pesticide intensity, grain yield, relative output rate, and integrated efficiency of each country based on multi-year average in 2002–2016.

Country Name	Pesticide Intensity (kg/hm^2^)	Cereal Yield (kg/hm^2^)	Pesticide Relative Output	Pesticide Integrated Efficiency
World	3.68	3562.97	0.59	0.68
France	3.74	7012.57	0.62	0.69
UK	13.16	7078.87	0.13	0.28
China	2.46	5500.07	0.85	0.28
US	9.82	6800.61	0.26	0.48
Netherlands	3.68	8191.63	0.59	0.68

**Table 4 ijerph-18-03771-t004:** Factor decomposition results for total amount of chemical fertiliser and pesticide used from 1997 to 2016, by LMDI.

Targets	1997–2006			2007–2016		
	ΔGDP-PI	ΔTPPPI-AP	ΔFLCI	ΔGDP-PI	ΔTPPPI-AP	ΔFLCI
Pesticides	69.73	−4.26	−12.10	126.31	80.08	19.14
Chemical fertiliser	35,694.64	−121.12	−344.30	3372.68	2138.40	511.19

GDP-PI: Value added of primary industry; TPPPI-AP: Agricultural producer price index; FLCI: Scale of rural operations.

**Table 5 ijerph-18-03771-t005:** Pesticide application intensity and pesticide output classifications of 31 provinces for 2016.

Category	Province
HI–HY	Guangdong, Liaoning, Zhejiang, Fujian, Jiangxi, Hubei
HI–LY	Hainan, Shanghai
LI–HY	Henan, Hebei, Jiangsu, Anhui, Shandong, Hunan, Guangxi, Sichuan
LI–LY	Ningxia, Beijing, Tianjin, Shanxi, Inner Mongolia, Jilin, Heilongjiang, Chongqing, Guizhou, Yunnan, Tibet, Shaanxi, Gansu, Qinghai, Xinjiang

**Table 6 ijerph-18-03771-t006:** Chemical fertiliser application intensity and fertiliser output categorisation for 31 Chinese provinces for 2016.

Categories	Province
HI–HY	Shandong, Jiangsu, Henan, Hubei, Guangdong
HI–LY	Beijing, Tianjin, Jilin, Shanghai, Fujian, Hainan, Shaanxi, Xinjiang
LI–HY	Sichuan, Hebei, Liaoning, Heilongjiang, Zhejiang, Anhui, Hunan, Guangxi
LI–LY	Qinghai, Shanxi, Inner Mongolia, Jiangxi, Chongqing, Guizhou, Yunnan, Tibet, Gansu, Ningxia

**Table 7 ijerph-18-03771-t007:** Categories of pesticide and fertiliser intensity and yield for the country’s 16 provinces for 2016.

Category	Province
HI–HY	Hubei, Guangdong
HI–LY	Hainan, Shaanxi
LI–HY	Sichuan, Hebei, Anhui, Hunan, Guangxi
LI–LY	Qinghai, Inner Mongolia, Guizhou, Yunnan, Tibet, Gansu, Ningxia

## Data Availability

Publicly available datasets were analyzed in this study. This data can be found here: [China Agricultural Statistics 1996–2016 http://www.fao.org/faostat/zh/#data] (accessed on 16 March 2021).

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
