# Peer review of "Evolution and Efficiency Assessment of Pesticide and Fertiliser Inputs to Cultivated Land in China"

_ijerph, 2021, doi:10.3390/ijerph18073771_

Round 1

Reviewer 1 Report

The requested corrections and clarifications have been made by the authors, so I suggest to be accepted this manuscript in present form.

Reviewer 2 Report

The authors addressed all of my concerns. 

This manuscript is a resubmission of an earlier submission. The following is a list of the peer review reports and author responses from that submission.

Round 1

Reviewer 1 Report

This work is dealing with the efficiency of fertilizer and pesticide utilisation in the light of yield and other agricultural outputs. The importance of this topic consists in the management and planning of optimal utilisation of pesticides and fertilizers in order to increase agricultural production keeping in mind the enivronmental load and human health also. In this way I think it is an important work. The introduction is quite welldone, the research design appropriate. The methods are also well described, but a little bit of simplification would make it more understandable for the ordinary readers who are not experts in this subject. The results are clearly presented in general, I have only two comments. 1/ fig. 8 is not clearly understandable, it needs revision and 2/ I suggest to put table 3 in appendix. The conclusions are fully supported by the results.

I'm not a big joker in English, but I found the text understandable, without serious mistakes, but maybe some simplification would be good to do for the better understanding.

Reviewer 2 Report

The topic studied by the authors is interesting and relevant. The authors did a comparative study of fertilizer and pesticide use intensity and integrated efficiency in China and four developed countries (UK, USA, France, and the Netherlands); identifies the driving factors and driving effects of increased pesticide and fertilizer inputs in China based on international discusses the exploratory effects of different provinces in reducing pesticide and fertilizer application and increasing efficiency in response to the results of the impact factor analysis. However, there are some major issues with the current version of the manuscript. For this, I recommend the major revision of the manuscript. Please, check the sentence structures, typos, and grammatical mistakes in the current version of the manuscript, and consider the following comments and revise the manuscript.

Abstract:

  • The conclusion is unclear.
  • The abstract should be a total of about 200 words maximum.
  • The abstract should be a single paragraph and should follow the style of structured abstracts, but without headings: 1) Background: Place the question addressed in a broad context and highlight the purpose of the study; 2) Methods: Describe briefly the main methods or treatments applied. Include any relevant preregistration numbers, and species and strains of any animals used. 3) Results: Summarize the article's main findings; and 4) Conclusion: Indicate the main conclusions or interpretations. The abstract should be an objective representation of the article: it must not contain results which are not presented and substantiated in the main text and should not exaggerate the main conclusions.

Introduction:

  • Add more information about the fertilizer and pesticide used in China (types, chemical characteristics and environmental impact).
  • Provide more current references (2016 – 2020).
  • L60 – L69: Add these lines into end of the introduction.
  • References must be numbered in order of appearance in the text (including table captions and figure legends) and listed individually at the end of the manuscript. In the text, reference numbers should be placed in square brackets [ ], and placed before the punctuation; for example [1], [1–3] or [1,3].
  • L70 – L81: Add these lines into Methodology

Materials and Methods:

  • Materials and methods be described with sufficient detail to allow others to replicate and build on published results. New methods and protocols should be described in detail while well-established methods can be briefly described and appropriately cited. Give the name and version of any software used and make clear whether computer code used is available.
  • L90-L151: Provides reference.
  • L188-L637: Here the authors combine methodology with results. They must separate these contents.
  • Line 129: [14]?
  • There are many formatting errors: use of superscript, messy text (equations), the equation should not be mentioned as “formula”, where (no Where and no indentation), etc.

Results and Discussions

  • Provide a concise and precise description of the experimental results, their interpretation as well as the experimental conclusions that can be drawn.
  • I think it is important that the authors report what has happened with their indicators from 2016 to date.

Conclusions

  • Authors are advised to provide some more details in the section “Conclusions.”

Reviewer 3 Report

Evolution and Efficiency Assessment of Pesticide and Fertilizer Inputs to Cultivated Land in China” considers the relative inputs/output of pesticides and fertilizer for 31 regions across China. Combining data from several databases, they find that the value-added of primary industry is the most significant predictor of pesticide and fertilizer outputs. I have outlined some concerns below. 1. I had a really hard time understanding what the overall research question they are trying to answer. a. On page 15, line 369 they state: “it is necessary to clarify the relationship between agricultural product prices and fertilizer use”. If this is what they are aiming for, I recommend moving this up to the introduction. b. Too much of the paper is spent comparing China to the rest of the world. This may be descriptively important, but is really a small piece of what they are trying to say (I think). The factor analysis seems to be a bigger contribution. I strongly recommend the authors decide on what their primary contribution is and build the paper around that. c. Similarly, the authors spend a significant portion of the paper discussing changes over time within China. These may be important points, but I was left wondering how they contributed to our understanding of the factors that drive fertilizer/pesticide efficiency 2. If the factor analysis is in fact their primary contribution, I would like to understand how they came up with those specific factors. It would seem availability of technology, as well as the quality of the soil would be important components. Perhaps these are incorporated into other measures but I didn’t see them. 3. I wasn’t clear on how they were calculating the “Value added of industry” measure. As this is seems to be the most important measure, the authors should describe this more thoroughly. I would also like to see some summary statistics regarding this measure in order to understand the variation. 4. On line 422 the authors discuss the contribution of each factor to CO2 emissions. As far as I can tell this is the only mention of CO2. Is this an artifact from a previous drat? I think maybe this should say total fertilizer use? 5. The structure of the paper, both in the writing and the organization, made following it extremely challenging.